# Pancreatic Ductal Cell-Derived Extracellular Vesicles Are Effective Drug Carriers to Enhance Paclitaxel’s Efficacy in Pancreatic Cancer Cells through Clathrin-Mediated Endocytosis

**DOI:** 10.3390/ijms23094773

**Published:** 2022-04-26

**Authors:** Haoyao Sun, Kritisha Bhandari, Stephanie Burrola, Jinchang Wu, Wei-Qun Ding

**Affiliations:** 1Department of Radiation Oncology, The Affiliated Suzhou Hospital of Nanjing Medical University, Suzhou 215001, China; sun464209459@gmail.com; 2Department of Pathology, University of Oklahoma Health Sciences Center, Oklahoma City, OK 73104, USA; kritisha-bhandari@ouhsc.edu (K.B.); stephaine-burrola@ouhsc.edu (S.B.)

**Keywords:** extracellular vesicles, pancreatic cancer, paclitaxel, clathrin, endocytosis

## Abstract

Chemo-resistance challenges the clinical management of pancreatic ductal adenocarcinoma (PDAC). A limited admittance of chemotherapeutics to PDAC tissues is a key obstacle in chemotherapy of the malignancy. An enhanced uptake of drugs into PDAC cells is required for a more effective treatment. Extracellular vesicles (EVs), especially small EVs (sEVs), have emerged as drug carriers for delivering chemotherapeutics due to their low immunogenicity and propensity for homing toward tumor cells. The present study evaluated sEVs derived from six different human cell lines as carriers for paclitaxel (PTX). The encapsulation of the chemotherapeutics was achieved using incubation, sonication and electroporation. The cytotoxicity of the EV drugs was evaluated by MTS assay. While sonication led to a higher efficiency of drug loading than incubation and electroporation, PTX loaded through incubation with HPNE-derived sEVs (HI-PTX) was the most efficacious in killing PDAC cells. Furthermore, HI-PTX was taken up by PDAC cells more efficiently than other EV drugs, implying that the efficacy of HI-PTX is associated with its efficient uptake. This was supported by the observation that the cytotoxicity and uptake of HI-PTX is mediated via the clathrin-dependent endocytosis. Our results indicate that the hTERT-HPNE cell-derived EVs are effective drug carriers to enhance paclitaxel’s efficacy in PDAC cells.

## 1. Introduction

Pancreatic ductal adenocarcinoma (PDAC) is a devastating disease, with its 5-year overall survival rate being less than 10% [1]. The high mortality rate of PDAC is related to the fact that the majority of pancreatic cancer patients have their tumor already metastasized at the time of diagnosis [2], making systemic therapy the mainstay of treatment. For those with advanced or metastasized tumors, chemotherapy is one of the most effective regimens recommended by the National Comprehensive Cancer Network guidelines [3]. Traditional chemotherapeutics are mainly small-molecule cytotoxic drugs that have distinct pharmacological profiles [4]. Upon intravenous administration, these drugs are passively distributed in the body via the bloodstream. Most of the drugs will gather in the liver and few can reach the tumor sites. Furthermore, drug distribution in the body mainly depends on the passive diffusion of the concentration gradient in the tumor microenvironment. It has been reported in the three-dimensional cell model that the penetration rate of small-molecule anti-tumor drugs in tumor tissue is only about 5% [5]. The limited admittance of chemotherapeutics to PDAC tissues in vivo is even more evident due to the unique stroma composition of PDAC that is histologically manifested as desmoplasia [6]. This pharmacological kinetic feature of chemotherapeutics causes traditional chemotherapy drugs to have serious side effects with limited potential of dose elevation in PDAC patients. 

To overcome this challenge, new chemo drug carriers have been developed to improve the pharmacokinetics, increase tumor-targeting efficacy and reduce the side effects of chemotherapeutics [7,8,9,10,11]. One of the recently studied natural Nano drug-carriers are extracellular vesicles (EVs), which are small lipid bilayer-delimited particles generated through various cellular processes and released from all types of cells investigated. They are able to transfer genetic and cellular materials between different cell types to mediate intercellular communication [12,13]. The most attractive properties of human cell-derived EVs as drug carriers include their lower immunogenicity and higher capacity of homing toward tumor cells [14,15,16,17]. Results from 12 recent clinical trials testing small EVs (sEVs) as therapeutic carriers or potential cancer therapeutics demonstrated safe profiles of sEVs delivered in humans, supporting the development of human cell-line-derived sEVs as chemotherapeutic carriers [18]. However, questions remain to be answered as to the choice of EV sources, the strategies of drug encapsulation, and the mechanisms of the cellular uptake of the delivered EV drugs. 

The present study was driven by the above-mentioned questions. We compared the drug loading efficiency of sEVs derived from six different human cell lines, including the human normal pancreatic duct cell line hTERT-HPNE, the human embryonic kidney cell line HEK-293T, the cancer-associated fibroblast cell line CAF19 and three human PDAC cell lines, PANC-1, MIA PaCa-2, and BxPC-3. Direct incubation, sonication and electroporation of sEVs were applied to incorporate paclitaxel (PTX) or gemcitabine (GEM), two commonly used chemotherapeutics. The sEV-drug efficacy was tested in the PDAC cell lines PANC-1, MIA PaCa-2, and BxPC-3. Our results show that while sonication leads to a higher efficiency of drug loading than incubation and electroporation, sEV derived from hTERT-HPNE cells and incubated with PTX (HI-PTX) was the most efficacious in killing PDAC cells. By using specific endocytosis pathway inhibitors and gene manipulation techniques, we demonstrated that the increased cellular cytotoxicity of HI-PTX is associated with enhanced cellular uptake of the sEV-drug complex through clathrin-mediated endocytosis.

## 2. Results

### 2.1. Characterization of sEVs and Drug Encapsulation

Small EVs were isolated from culture medium of the six human cell lines as we described [19]. Protein analysis of isolated sEVs showed that hTERT-HPNE sEV has the lowest protein concentration, while MIA PaCa-2 sEV the highest. Cancer-cell-derived sEVs seemed to have a higher EV concentration than normal-cell-derived sEVs (Figure 1a). EV particle quantity derived from the six cell lines also differed, but not as much as the protein concentrations (Figure 1b). Two sEV-positive markers, flotillin-1 and CD63, and one negative marker, calnexin, were detected by Western blot to verify the isolated sEVs (Figure 1c). All sEVs had iconic peaks around 100–200 nm, based on nanoparticle tracking analysis (Figure 1d). To quantify EV-drug concentration, both EV drugs and sEVs were analyzed by Nanodrop in the UV–Vis spectrum. Standard curves of free PTX, GEM and sEVs derived from the six cell lines were generated separately at 230 or 275 nm (Appendix A). The quantity of loaded drugs was expressed as ng of drug/μg of sEV. Our data showed that sonication leads to a higher concentration of drug loading than incubation and electroporation (Figure 1e).

### 2.2. HPNE sEV-PTX Derived from Incubation (HI-PTX) Is most Efficacious in Killing Pancreatic Cancer Cells

To test whether the EV-PTX are cytotoxic toward cancer cells, PANC-1, MIA PaCa-2 and BxPC-3 cells were seeded in 96-well plates and treated with free or EV drugs for 24–72 h. MTS assay was applied to assess cell cytotoxicity as we previously described [20,21]. As expected, PTX and GEM suppressed cell viability in a time- and concentration-dependent manner, with IC_50_ values ranging from 10 to 100 nM in the three cancer cell lines (data not shown). We then treated the cells using EV drugs or free drugs with the same drug concentrations ranging from 1 to 1000 nM for 72 h. While five EV-PTX drugs showed lower IC_50_ values than that of free PTX, only HI-PTX showed consistent cytotoxicity in all three cancer cell lines (Figure 2a–f). In contrast to EV-PTX, by following the same protocol and procedures, no EV-GEM drugs were more cytotoxic than free GEM in PANC-1 cells (Figure 2g,h). These results indicate that the efficacy of EV-encapsulated chemotherapeutics is associated with the loading methods, EV sources, and the drug of interest. 

### 2.3. Uptake of HI-PTX in PANC-1 Cells

To understand why cytotoxicity differs among the EV drugs, we examined EV-drug uptake using the PKH67 dye under a fluorescent microscope. We found that both HPNE sEVs or HI-PTX uptake by PANC-1 cells were time-dependent, with the highest uptake at 10 h post sEV addition. Interestingly, HI-PTX uptake was more pronounced than HPNE sEV uptake at each time point (Figure 3a). To exclude the potential effects of PTX on fluorescent imaging, we compared the uptake of HPNE sEV versus HPNE sEV plus free PTX and found no differences in any of the three cancer cell lines between the two groups of sEVs (Figure 3b). To determine whether the method of drug encapsulation affects the uptake, we compared the uptake of sEVs derived from HPNE and HEK-293T cells and prepared with PTX via sonication, electroporation, and incubation. It turned out that HI-PTX had the highest uptake of the EV drugs in PANC-1 and BxPC-3 cells (Figure 3c), consistent with its more pronounced cytotoxicity. While there was also a tendency for increased uptake of HI-PTX in MIA PaCa-2 cells, statistical significance could not be reached. These results suggest a connection between cell cytotoxicity and uptake of the EV drugs.

### 2.4. HI-PTX’S Uptake and Cytotoxicity Is Associated with Clathrin-Mediated Endocytosis

To explore the mechanism of HI-PTX uptake and cytotoxicity in pancreatic cancer cells, several endocytic pathway inhibitors were applied. The higher uptake of HI-PTX was diminished by the inhibitors Monesin, Bafilomycin A1 (BFA), and clathrin-mediated endocytosis inhibitor Pitstop2. To the contrary, the caveolin-mediated endocytosis inhibitor Genistein had no effect on HI-PTX uptake (Figure 4a). However, among the inhibitors tested, only Pitstop2 could reverse the cytotoxicity of HI-PTX in PANC-1 cells when 3 nM PTX equivalent concentration of HI-PTX was applied (Figure 4b), indicating that the clathrin-mediated endocytosis is primarily involved in HI-PTX’s uptake and cytotoxicity. To further confirm the contribution of clathrin-mediated endocytosis to HI-PTX uptake, overexpression and siRNA knockdown of clathrin light chain and caveolin was achieved in PANC-1 cells (Figure 4c,d). The most successful knockdown of clathrin light chain was obtained using Si-CLTB-93, which was used for subsequent experiments (Figure 4e,f). As shown in Figure 4e, EV uptake was associated with the expression levels of clathrin, not that of caveolin, for both HPNE sEVs and the HI-PTX. These observations support the conclusion that HI-PTX uptake is facilitated, at least in part, by clathrin-mediated endocytosis in pancreatic cancer cells. To make sure the knockdown of clathrin light chain impairs clathrin-mediated endocytosis, an RFP-tagged transferrin receptor construct (TfR-pHuji plasmid [22] (Addgene Plasmid #61505) was used to monitor cellular localization of the transferrin receptor during siRNA knockdown (Appendix A). It confirmed that the knockdown leads to more transferrin receptor on the cell surface. This is consistent with previous reports showing that the depletion of clathrin light chain effectively inhibits the clathrin-mediated internalization of cargos such as bacteria and virus particles that are too large for conventional endocytosis [23,24] and that exosomes or small EVs share physical properties and size ranges with viral particles [25]. 

## 3. Discussion

Although the experimental evidence showing the effectiveness of EV drugs against cancer has been abundant, there has been no clear consensus regarding the choice of methods for EV encapsulation of drugs, the source of EVs as carriers, and the mechanisms of EV-drug internalization to achieve the best therapeutic effects. The results from the present study demonstrate that, while sonication leads to higher efficiency of PTX loading than incubation and electroporation, HI-PTX prepared by incubation is most efficacious in killing pancreatic cancer cells, observations in line with a recent report using EV-encapsulated doxorubicin (DOX) [26]. Furthermore, we demonstrated that the uptake and cytotoxicity of HI-PTX are associated with clathrin-mediated endocytosis in pancreatic cancer cells, implicating endocytosis pathways in EV-drug efficacy. 

Cellular uptake of molecules larger than one kilo Dalton, such as proteins or nanoparticles, is usually facilitated by endocytic pathways [4]. It has been reported that EV uptake is mediated through various endocytic pathways, including clathrin-dependent endocytosis and clathrin-independent endocytosis, such as caveolin-mediated uptake and lipid raft-mediated internalization. Because EV populations are often heterogeneous, more than one route of uptake is generally involved during EVs internalization into cells [27]. For example, clathrin- and caveolin-dependent endocytosis and macropinocytosis are the predominant routes of sEV-mediated communication between bone marrow stromal cells and multiple myeloma cells, and the knocking down of calveolin-1 and clathrin heavy chain in multiple myeloma cells significantly suppressed sEV uptake and chemo sensitivity to bortezomib [28]. However, the endocytosis pathways involved in the uptake of drug-loaded EVs has not been previously established. Our experiment results showed that the cellular uptake of HI-PTX (PTX encapsulated by HPNE sEVs via incubation) is enhanced when compared to the uptake of HPNE sEVs in all three pancreatic cancer cell lines. This enhanced uptake of HI-PTX is attributed to clathrin-mediated endocytosis, since modulation of this process using the inhibitor Pitstop2 or by expression manipulation of clathrin altered the uptake of HI-PTX and its cytotoxicity, whereas the caveolin-dependent endocytosis seemed to be irrelevant in this process. Our observations thus provide novel information in the understanding of EV drug uptake and efficacy in cancer cells. The higher uptake of HI-PTX, when compared to EV-PTX prepared via sonication and electroporation, may be explained by the possibility that, compared to incubation, both sonication and electroporation are likely to trigger a harsh process to sEVs that cause damage of sEV membranes, thereby compromising their ability for cellular internalization [29]. Nonetheless, the mechanisms responsible for a higher uptake of HI-PTX, when compared to the uptake of HPNE sEVs, remain to be explored in the future.

Various sources of sEVs have been tested for their potential as therapeutic carriers, and each type of sEV may have pros and cons when used for drug delivery [30]. This is most likely due to their differences in the EV cargos in which each may have unique lipid, protein and RNA profiles that may directly influence sEVs’ ability to interact with receiving cells [31]. In particular, tumor-cell-derived EVs were considered drug carriers for selective targeting and enhanced immune response, yet may ironically promote tumor growth and invasion, due to their cargo compositions [30]. In this context, a strategy using exosomes coated with magnetic nanoparticles to deliver chemotherapeutics specifically to tumor cells has been described [32,33,34]. However, the best EV sources have yet to be identified in the development of EVs as therapeutic carriers. In this study, we tested sEVs derived from six human cell lines, including cancer lines and non-cancer lines. We found that the sEVs derived from the human pancreatic ductal cell line HPNE (a non-cancer line) are the most efficacious when used to encapsulate PTX via incubation, suggesting that this group of sEVs is a promising candidate for further development as cancer therapeutic carriers. In vivo testing is warranted for the efficacy and safety of HI-PTX. 

An interesting finding from this study was that in contrast to HI-PTX, no EV-GEM drugs showed superior cytotoxicity compared with free GEM in pancreatic cancer cells, suggesting that individual chemotherapeutics entails a different fate when encapsulated by sEVs. Specific to GEM, two research groups have reported that GEM is successfully loaded into autologous exosomes that suppress tumor growth [35,36]; however, others have also reported that GEM is inefficiently entrapped by nanoparticles due to leakage or hydrophilic property [37,38]. These inconsistent observations, along with ours, are most likely due to the nature of the sEVs used, and the procedures applied for sEV isolation and drug encapsulation. 

In summary, we have demonstrated that HI-PTX is the most efficacious EV-PTX in suppressing pancreatic cancer cell viability, which is primarily mediated via the clathrin-dependent endocytosis pathway. Our data indicate that the efficacy of EV-encapsulated chemotherapeutics is associated with the loading methods, EV sources, and EV uptake efficiency.

## 4. Materials and Methods

Cell culture. The human pancreatic cancer cell lines PANC-1, MIA PaCa-2 and BxPC-3, the immortalized human pancreatic duct cell line HPNE, and the human embryonic kidney cell line HEK-293T were obtained from the American Type Culture Collection (ATCC, Manassas, VA, USA). The cancer-associated fibroblast cell line CAF19 was kindly provided by Dr. Priyabrata Mukherjee, University of Oklahoma Health Sciences Center. Cells were cultured following ATCC’s instructions, and CAF19 was cultured in DMEM supplemented with 10% FBS. Exosome-depleted FBS and horse serum were prepared by pelleting the serum EVs at 100,000× *g* for 1.5 h at 4 °C and then filtered through a 0.22 µm PVDF centrifuge filter. Cells were routinely incubated in a humidified environment at 37 °C and 5% CO_2_.

EV isolation and validation. EVs from 20 mL cell culture medium were isolated following the protocol we previously described, with minor modifications [19,39]. Briefly, after being pre-cleared by 10,000× *g* centrifugation for 30 min at 4 °C, the resulting supernatant was transferred to a 100 kDa cut-off centrifugal column (Merckmillipore, Burlington, MA, USA) and centrifuged for 15 min at 4000× *g*. The concentrated supernatant was then filtered through PVDF centrifuge filters (Merckmillipore, Burlington, MA, USA) as we described [39]. Small EVs (exosomes) were recovered using the Total Exosome Isolation Reagent (Thermofisher, Waltham, MA, USA) following the manufacturer’s instructions. The isolated sEVs dissolved in PBS were verified by Western blot detecting positive and negative exosome proteins and nanoparticle tracking analysis (Nanosight NS300 System, Malvern Instruments, Malvern, UK) measuring sEV sizes and concentrations. 

Drug encapsulation and quantification. PTX (Sigma-Aldrich, St. Louis, MO, USA, 0.01 µmol, CAS 33069-62-4) or GEM (Sigma-Aldrich, St. Louis, MO, USA, 0.1 µmol, CAS 122111-03-9) was mixed with the purified sEVs (around 50 µg) in 1 mL PBS. Three loading methods, including incubation, sonication and electroporation, were applied. For the incubation method, the sEV-drug mixture was incubated at 37 °C for 1 h. For the sonication method, the mixture was sonicated using a FB505 sonicator (Fisher Scientific, Pittsburgh, PA, USA) with the following settings: 20% amplitude, 6 cycles of 30 s on and off, followed by 2 min cooling period. After sonication, the EV-drug mixture was incubated at 37 °C for 1 h. For the electroporation method, the sEV-drug mixture was electroporated using the P3 Primary Cell 4D-NucleofectorTM X Kit L (Lonza, Basel, Switzerland) with DN-100 program of the 4D-NucleofectorTM Core Unit. After electroporation, the EV-drug mixture was also incubated at 37 °C for 1 h. The EV-drug mixture was then pelleted by the total exosome isolation reagent and dissolved in 200 μL PBS to remove unbounded drugs. UV absorbance of the EV-drug mixtures at 230 nm (PTX) or 275 nm (GEM) was recorded by Nanodrop (Denovix, Wilmington, DE, USA) to determine the drug concentrations as previously described [40,41]. A standard curve of free PTX or GEM was established, and drug concentration in the EV-drug mixture was calculated as the following: EV-drug UV absorbance minus sEV UV absorbance, and the resulting absorbance was fitting to the free drug standard curve. The drug loading efficiency was presented as drugs (ng)/sEVs (μg).

Western blot analysis. Western blot was performed as we recently described [39]. Primary antibody raised against Clathrin-LC was obtained from Santa Cruz Biotechnology (Dallas, TX, USA), and those against Caveolin 1 and GAPDH were purchased from Cell Signaling Technology (Danvers, MA, USA). Antibodies used for sEV marker detection include: CD63 (Santa Cruz, Dallas, TX, USA), Flotillin-1 and Calnexin (Cell Signaling Technology, Danvers, MA, USA). The Li-Cor Odyssey^®^ Fc Imaging System was used to visualize and image the blots.

Cell proliferation (MTS) assay. Cells were seeded onto 96-well plates at a density of 6000–8000 cells/well in triplicate and drugs were added the next morning. After 72 h incubation at 37 °C with 5% CO_2_, the medium was replaced with 100 μL fresh medium supplemented with 20 μL CellTiter 96^®^ AQueous One Solution (Promega, Madison, WI, USA). After 1 h of incubation, the absorbance value at 490 nm was recorded using a spectrometer (VWR SpectraMax^®^ 190, Radnor, PA, USA). The data were expressed as percentage of the absorbance value detected in untreated control cells. 

EV uptake analysis. Small EVs were labelled using the PKH67 Green Fluorescent Cell Linker kit (Sigma-Aldrich, St. Louis, MO, USA) following the manufacture’s protocol. Same amount of sEV or EV-drug (around 20 μg) was added to 500 μL diluent before adding 1 μL PKH67. The mixture was incubated at room temperature for 20 min and 500 μL EV-depleted FBS was added to stop the labeling. The labeled sEVs were pelleted by ultracentrifuge at 100,000× *g*, 4 °C for 1.5 h. The pellet was resolved in EV-depleted medium. Cancer cells (5 × 10^4^ cells/well) were plated on a 24-well plate. The labeled sEVs or sEV-drug mixtures (around 20 μg) were added to the cell culture 24 h post seeding and incubated for 2 to 10 h. In some experiments, endocytosis inhibitors were added 6 h prior to addition of the labeled sEVs or EV drugs. Cells were washed by PBS, fixed by paraformaldehyde and mounted using Prolong Antifade Reagents (Thermofisher, Waltham, MA, USA). Fluorescent signal in cells was detected by a fluorescent microscope (Thermofisher, Waltham, MA, USA) and analyzed using the ImageJ software [42].

Manipulation of clathrin and caveolin expression. The mCherry-clathrin-LC and mCherry-caveolin1 plasmids were purchased from Addgene (Watertown, MA, USA). The mCherry-C1 was purchased from Takara Bio USA (Mountain View, CA, USA) as control. DNA transfection to PANC-1 cells was performed using Lipofectamine 3000 (Fisher Scientific, Pittsburgh, PA, USA) and clathrin, and caveolin overexpression was verified by fluorescent microscopy and Western blot. For clathrin light chain and caveolin knockdown, three clathrin siRNA—s3191 (siCLTB91), s3192 (siCLTB92), and s3193 (siCLTB93)—one caveolin siRNA—s2446 (siCAV1)—and one negative control siRNA were synthesized (Table 1, Thermofisher, Waltham, MA, USA). Transfection of the siRNAs to PANC-1 cells was performed using Lipofectamine 3000 (Fisher Scientific, Pittsburgh, PA, USA) and the knockdown of clathrin and caveolin was verified by Western blot.

## Figures and Tables

**Figure 1 ijms-23-04773-f001:**
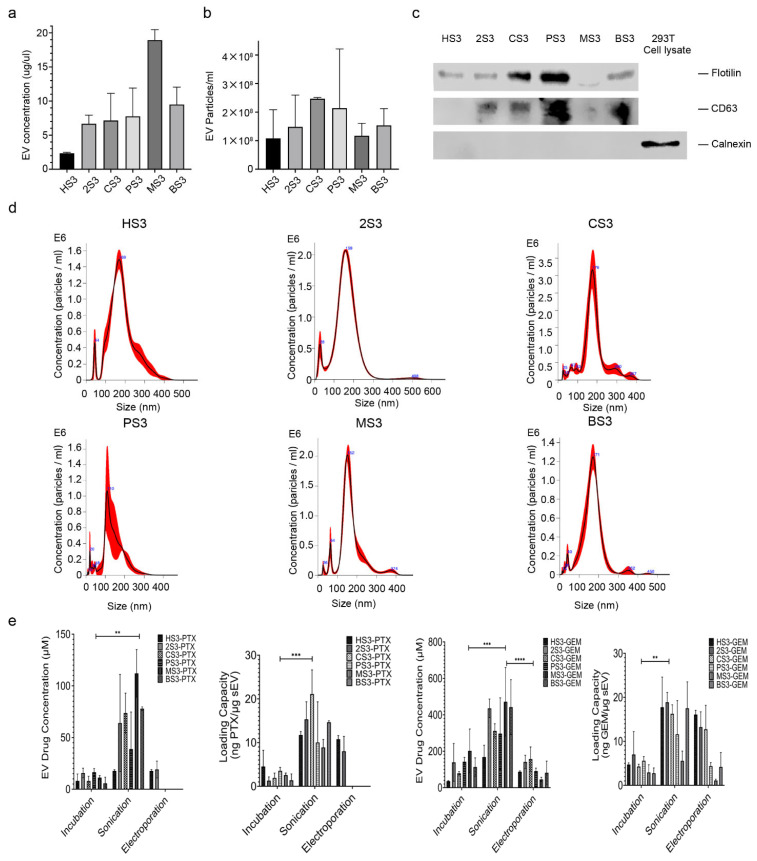
Validation of extracellular vesicles and quantification of EV drugs. (**a**) EVs were derived from 20 mL medium of 6 different human cell lines, and EV concentrations were analyzed by BCA assay; (**b**) EV particle analysis by NanoSight (1:1000 dilution); (**c**) Small EV-positive markers, CD63 and Flotillin-1, and a negative marker, Calnexin, detected by Western blot; (**d**) Representative EV size distribution and particle numbers analyzed by Nanosight; (**e**) Sonication leads to higher efficiency of EV drug loading than incubation and electroporation. **** *p* < 0.0001, *** *p* < 0.001, ** *p* < 0.01, one-way ANOVA (*n* = 3–5, comparison of Incubation, Sonication, and Electroporation). HS3: HPNE sEV; 2S3: HEK-293T sEV; CS3: CAF19 sEV; PS3: PANC-1 sEV; MS3: MIA PaCa-2 sEV; BS3: BxPC-3 sEV; PTX: Paclitaxel; GEM: Gemcitabine.

**Figure 2 ijms-23-04773-f002:**
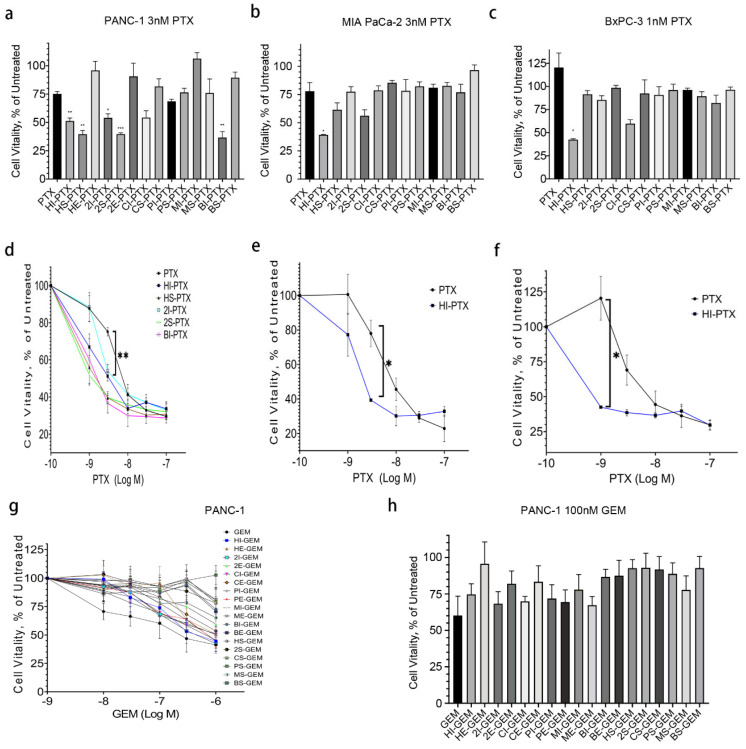
HI-PTX is more cytotoxic than other EV drugs. (**a**,**d**) Cell vitality of PANC−1 cells treated with equivalent of 3 nM PTX and 1 to 100 nM EV−PTX for 72 h; (**b**,**e**) Cell vitality of MIA PaCa−2 cells treated with equivalent of 3 nM PTX and 1 to 100 nM EV−PTX for 72 h; (**c**,**f**) Cell vitality in BxPC−3 cells treated with equivalent of 1 nM PTX or 1 to 100 nM EV−PTX for 72 h; (**g**,**h**) Cell vitality of PANC−1 cells treated with equivalent of 100 nM GEM or 10 nM to 1 µM EV−GEM for 72 h. The cytotoxicity of all EV−GEM was lower than that of free GEM. Statistical analysis was performed using one-way ANOVA followed by Dunnett’s post-test for (**a**) and two-way ANOVA for (**d**–**f**). *** *p* < 0.001, ** *p* < 0.01, * *p* < 0.05 (data from three individual determinations). HI-PTX: Incubation of HPNE sEV with paclitaxel; HS−PTX: Sonication of HPNE sEV with paclitaxel; HE−PTX: Electroporation of HPNE sEV with paclitaxel; 2I-PTX: Incubation of HEK−293T sEV with paclitaxel; 2S−PTX: Sonication of HEK−293T sEV with paclitaxel; 2E−PTX: Electroporation of HEK−293T sEV with paclitaxel; CI-PTX: Incubation of CAF19 sEV with paclitaxel; CS−PTX: Sonication of CAF19 sEV with paclitaxel; PI−PTX: Incubation of PANC−1 sEV with paclitaxel; PS−PTX: Sonication of PANC−1 sEV with paclitaxel; MI−PTX: Incubation of MIA PaCa−2 sEV with paclitaxel; MS−PTX: Sonication of MIA PaCa−2 sEV with paclitaxel; BI−PTX: Incubation of BxPC−3 sEV with paclitaxel; BS−PTX: Sonication of BxPC−3 sEV with paclitaxel.

**Figure 3 ijms-23-04773-f003:**
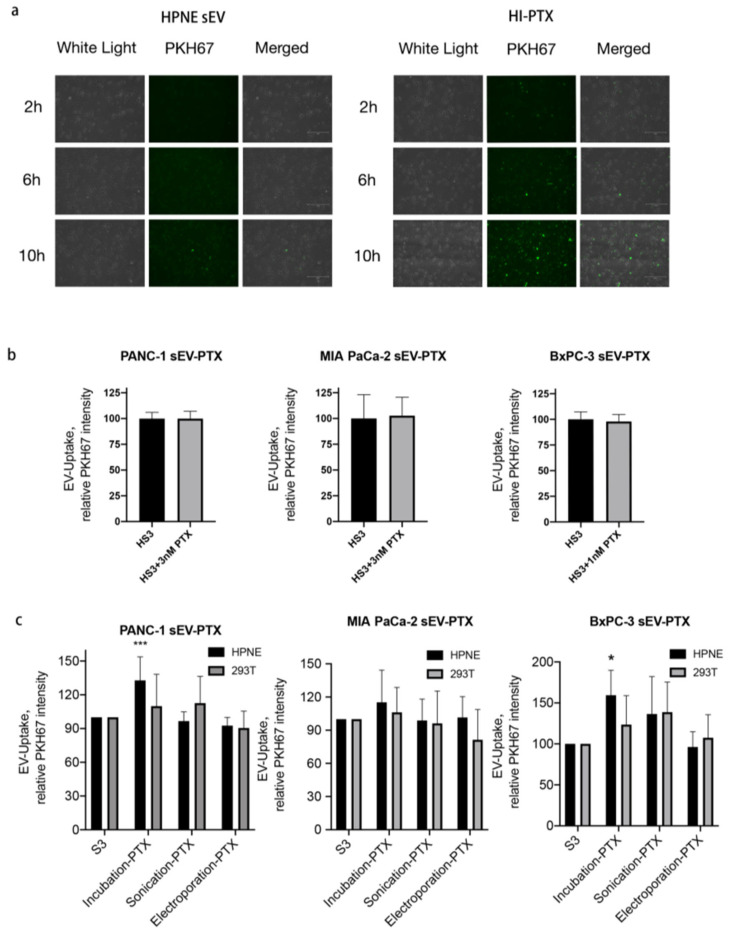
HI-PTX is taken up more effectively by pancreatic cancer cells. (**a**) Equal amounts of HPNE-sEV and HI-PTX were labeled with PKH67 and loaded onto PANC-1 cells for 2 to 10 h. Cellular uptake of HPNE-sEV and HI-PTX was monitored by fluorescent microscopy (excitation 488 nm, detection, 510 nm; Nikon TE2000-E, 10× magnification); (**b**) Adding HPNE-sEV and PTX simultaneously to PANC-1, MIA PaCa-2, and BxPC-3 cells did not enhance EV uptake. The fluorescence was detected by fluorescent microscopy and the fluorescence intensity was semi quantified, and presented as relative levels; (**c**) The uptake of HPNE- and 293T-derived EV, when loaded with PTX by incubation, sonication and electroporation, in the three pancreatic cancer cell lines. *** *p* < 0.001, * *p* < 0.05, one-way ANOVA followed by Dunnett’s post-test (data from three individual determinations).

**Figure 4 ijms-23-04773-f004:**
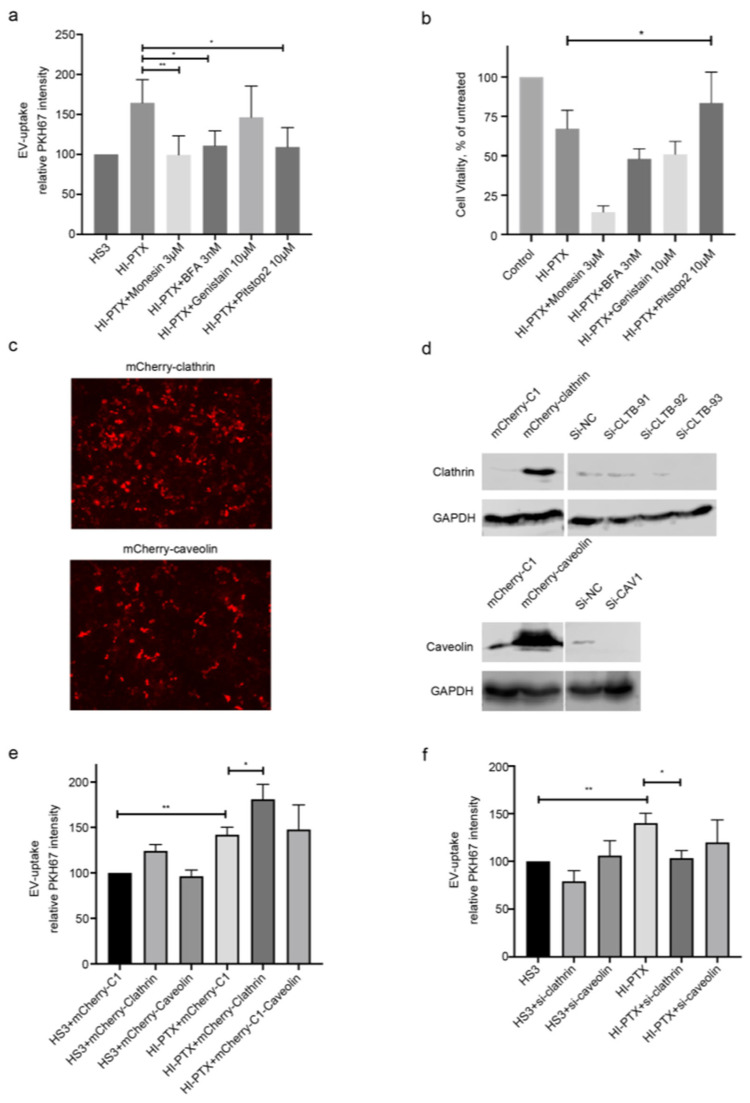
HI-PTX’s uptake and cytotoxicity is associated with clathrin-mediated endocytosis. (**a**) The clathrin-mediated endocytosis inhibitor Pitstop2 inhibited the uptake of HI-PTX by PANC-1 cells; (**b**) Treatment with Pitstop2 attenuated HI-PTX’s cytotoxicity in PANC-1 cells; (**c**) Overexpression of clathrin and caveolin (Nikon TE2000-E, 10× magnification); (**d**) Knockdown of clathrin and caveolin expression in PANC-1 cells, evidenced by fluorescent images and Western blot analysis, mCherry-clathrin, 63 kDa; clathrin, 31 kDa; mCherry-caveolin, 53 kDa; caveolin, 22 kDa; (**e**,**f**) The uptake efficiency of HI-PTX was altered by overexpression or knockdown of clathrin in pancreatic cancer cells. ** *p* < 0.01, * *p* < 0.05, one-way ANOVA followed by Dunnett’s post-test (data from three individual determinations).

**Table 1 ijms-23-04773-t001:** siRNA sequences for knockdown of clathrin light chain and caveolin.

Name	Sequence	Target
si_CLTB91	sense: GCCCAGCUAUGUGACUUCATT	
	antisense: UGAAGUCACAUAGCUGGGCCA	Clathrin light chain
si_CLTB92	sense: CCUCCUCUCAGUCUACUCATT	
	antisense: UGAGUAGACUGAGAGGAGGCG	Clathrin light chain
si_CLTB93	sense: GAACAAGUAGAGAAGAACATT	
	antisense: UGUUCUUCUCUACUUGUUCAC	Clathrin light chain
si_CAV1	sense: GCUUCCUGAUUGAGAUUCATT	
	antisense: UGAAUCUCAAUCAGGAAGCTC	Caveolin

Statistics. Statistical analyses were performed using GraphPad Prism software (La Jolla, CA, USA). One-way ANOVA was used to determine *p*-values among experimental groups and a *p*-value of ≤0.05 was considered statistically significant.

## Data Availability

The data presented in this study are contained within the article and Appendix A.

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
