# Peer review of "Pancreatic Ductal Cell-Derived Extracellular Vesicles Are Effective Drug Carriers to Enhance Paclitaxel’s Efficacy in Pancreatic Cancer Cells through Clathrin-Mediated Endocytosis"

_ijms, 2022, doi:10.3390/ijms23094773_

Round 1
Reviewer 1 Report
This present article by Sun et al discusses improvement of treatment challenges in pancreatic ductal carcinoma by increasing efficiency of drug encapsulation. I am in principle supportive of accepting this work for publication only after the following improvements were made in its present form. However, I have few suggestions to improve the manuscript for publication.
Major:
- Fig 3, author quantified drug uptake from 3 different cell lines, however, only one was presented. At least the other two images should be included in the supplementary.
- In supplementary 2, the author validated siRNA clathrin using TfnR receptor uptake of Tfn. However, I can not see any notable difference between control and siRNA treated sample. I suggest using the following approach from a published paper from Mani et al Mol Biol Cell . 2016 Jan 15;27(2):334-48 to redesign the experimental approach on target validation.
Minor comments:
- x-y axis in fig 1 panel d, should be the same in order to compare size distribution between the samples.
- In fig 2, the graph legends panel d-g were not legible, and should be re-size.
- In fig3, panel a and also in fig4 panel c scale bar and magnification was not indicated.
- In fig 3 panel c p-value missing in MIA PcCa-2 sEV-PTX, it should be documented in the graph.
Author Response
Major
1. Fig 3, author quantified drug uptake from 3 different cell lines, however, only one was presented. At least the other two images should be included in the supplementary.
Response: We have included images showing the drug uptake by the other two cell lines (Supplemental_Figure 3).
2. In supplementary 2, the author validated siRNA clathrin using TfnR receptor uptake of Tfn. However, I can not see any notable difference between control and siRNA treated sample. I suggest using the following approach from a published paper from Mani et al Mol Biol Cell . 2016 Jan 15;27(2):334-48 to redesign the experimental approach on target validation.
Response: Thanks for the suggestions. We have now updated supplementary figure 2 to include quantified fluorescent intensities (please see the updated Supplemental Figure 2).
Minor
1. x-y axis in fig 1 panel d, should be the same in order to compare size distribution between the samples.
Response: Thanks for the comment. The x-y in figure 1 is automatically arranged by the Nanosight NS300 System.
2. In fig 2, the graph legends panel d-g were not legible, and should be re-size.
Response: We have resized the legends for panel d-g in figure 2 in the revision.
3. In fig3, panel a and also in fig4 panel c scale bar and magnification was not indicated.
Response: We have included the magnification for these panels in the figure legends in the revision.
4. In fig 3 panel c p-value missing in MIA PcCa-2 sEV-PTX, it should be documented in the graph.
Response: Although, as shown in figure 3c, there is a tendency that more sEV-PTX is taking up by MIA PaCa-2 cells via incubation, statistical difference is not achieved.
Reviewer 2 Report
The author's investigation on the choice EV sources, the strategies of drug encapsulation, and the mechanisms of cellular uptake of the delivered EV drugs are well designed with significant scientific problems proposed, which is a promising study in the direction of EVs based drug developments. Thus, this paper deserves publication after addressing the following minor points.
- Figure 3A is very vague and should be replaced with a high dpi picture.
- Significant differences should be analyzed in Figure 1E and Figure 2.
- In recent years, various exosome-based therapeutic drug delivery system has been proposed such as Biomaterials 276, 121056, Advanced Functional Materials 28 (18), 1707360, ACS applied materials & interfaces 9 (33), 27441-27452. The authors are suggested to discuss these works as well.
- In the Experimental section the authors are suggested adding CAS numbers for all chemicals to do this section clearer and more accurate.
Author Response
1. Figure 3A is very vague and should be replaced with a high dpi picture.
Response: the vague of Figure 3A is largely caused by the small scale pf the panels, which is arranged to accommodate the whole Figure 3. We have now included a supplemental-Figure 4 to show the resolution of the Figure 3A.
2. Significant differences should be analyzed in Figure 1E and Figure 2.
Response: We have performed statistical analysis of the Figure 1E and Figure 2. The figures and figure legends are updated in the revision.
3. In recent years, various exosome-based therapeutic drug delivery system has been proposed such as Biomaterials 276, 121056, Advanced Functional Materials 28 (18), 1707360, ACS applied materials & interfaces 9 (33), 27441-27452. The authors are suggested to discuss these works as well.
Response: We have discussed these reports and added these references to the revision (line 194-195, page 9).
4. In the Experimental section the authors are suggested adding CAS numbers for all chemicals to do this section clearer and more accurate.
Response: We have added the CAS numbers for the key compounds used in this study.